# Disturbance modifies payoffs in the explore-exploit trade-off

Shay O'Farrell[1], James N. Sanchirico[1,2], Orr Spiegel [3], Maxime Depalle[1], Alan C. Haynie[4], Steven A. Murawski[5], Larry Perruso[6] & Andrew Strelcheck[7]

Decision-making agents face a fundamental trade-off between exploring new opportunities with risky outcomes versus exploiting familiar options with more certain but potentially suboptimal outcomes. Although mediation of this trade-off is essential to adaptive behavior and has for decades been assumed to modulate performance, the empirical consequences of human exploratory strategies are unknown beyond laboratory or theoretical settings. Leveraging 540,000 vessel position records from 2494 commercial fishing trips along with corresponding revenues, here we find that during undisturbed conditions, there was no relationship between exploration and performance, contrary to theoretical predictions. However, during a major disturbance event which closed the most-utilized fishing grounds, explorers benefited significantly from less-impacted revenues and were also more likely to continue fishing. We conclude that in stochastic natural systems characterized by non-stationary rewards, the role of exploration in buffering against disturbance may be greater than previously thought in humans.

[1] Department of Environmental Science and Policy, University of California Davis, One Shields Avenue, Davis, CA 95616, USA. [2] Resources for the Future, Washington, DC 20036, USA. [3] School of Zoology, Faculty of Life Sciences, Tel Aviv University, Tel Aviv 69978, Israel. [4] NOAA Fisheries, Alaska Fisheries Science Center, 7600 Sand Point Way NE, Bldg 4, Seattle, WA 98115, USA. [5] College of Marine Science, University of South Florida, 140 Seventh Avenue South, MSL 200D, St. Petersburg, FL 33701, USA. [6] NOAA Fisheries, Southeast Fisheries Science Center, 75 Virginia Beach Dr., Miami, FL 33149, USA. [7] NOAA Fisheries, Southeast Regional Office, 263 13th Avenue South, St. Petersburg, FL 33701, USA. Correspondence and requests for materials should be addressed to S.O. (email: sofarrell@ucdavis.edu)

Exploration provides us with information about the surrounding world[1–3]. Whenever we take a new route to work, for example, we are sampling our environment and adding to a store of information that may increase our long-term benefits (e.g., finding the fastest commute) and/or confer resilience to system dynamics or disturbance should present options become less attractive or unavailable (e.g., knowing alternative routes if traffic is heavy). However, exploration comes at a cost, as it involves increased investment with uncertain outcomes, and time and resources could instead have been invested in exploiting current knowledge to gain immediate, tangible benefits[2,4,5] This explore/exploit trade-off (EETO) is pervasive in sequential decision-making settings from financial portfolio blending to machine learning to animal foraging[6–10], with agents displaying EETO-mediating strategies that place varying emphasis on exploration. However, despite the importance of EETO mediation to adaptive behavior in complex environments[2,11], many normative and empirical aspects of the problem are poorly understood[2].

Behavioral researchers commonly investigate EETO mediation using bandit tasks, where the subject plays a series of one-armed-bandit machines or analogous devices with the goal of devising an EETO strategy with the highest aggregate payoff[5]. At each choice occasion, the subject decides whether to play the same 'machine' or move to another. For example, a subject who eschews exploration could waste time exploiting a machine that only pays off once every 10 trials without ever discovering that a neighboring machine pays off once every three trials. Bandit tasks have traditionally assumed that the probability of payoff from a given option (i.e., a machine) is stationary and that the portfolio of options (i.e., the array of machines) from which the subject chooses remains constant[2,11]. Although progress has been made in relaxing the assumption of reward stationarity[12], investigation of EETO in the lab remains a profoundly different decision-making setting from the natural systems to which the human EETO-mediating apparatus is adapted[5,13]. Natural systems are subject to ecological and environmental fluctuations which stochastically modify both the payoff probability of a given option, as well as the portfolio of available options. Under these circumstances, the consequences of operating at various positions along the continuum of EETO strategies remain untested.

An ideal system for investigating EETO strategies in a natural setting is commercial fishing, one of our last remaining hunter–gatherer activities[13]. First, vessel captains must repeatedly decide whether to exploit previously sampled fishing grounds whose quality is known or to explore new locations. Second, the payoff of fishing grounds fluctuates within and across years and the portfolio of options changes over time due to both regulatory rules (e.g., seasonal closures) and environmental variability (e.g., rough seas). Third, commercial fisheries present one of the few social–ecological systems where there exist both spatiotemporal data on behavior (vessel movement tracks) and explicit payoffs (revenue).

We assess real-world payoffs of contrasting EETO strategies using longline fishing in the US Gulf of Mexico (GoM), leveraging a unique natural experiment where a large portion of prime fishing grounds was closed for five months in 2009. The closure stemmed from excessive bycatch of endangered species of sea turtles[14]. As a result of the emergency closure, many affected vessels were forced to relinquish habitually fished grounds and either fish elsewhere or retire during the closure. As the intervention was a fisheries management action rather than an emergency closure, such as an oil spill, no compensation was given to the longline fleet.

Given that exploration and exploitation are considered to trade-off against each other in the EETO framework[5], we assume that strategies investing too heavily towards either end of the spectrum are likely to be suboptimal and that before the disturbance, vessel-level performance may be maximized around intermediate strategies that more evenly balance exploration and exploitation. During the disturbance, however, we hypothesize that more-exploratory strategies should buffer against adverse impacts, when vessels with diversified portfolios of fishing grounds should benefit from their enhanced knowledge of non-impacted resources.

To capture contrasting aspects of exploratory behavior, we use two metrics: patch residence time (PRT) and choice entropy. These metrics focus, respectively, on each of two nested fishing decisions: (1) should the vessel continue fishing at the current location or move on; and (2) if the vessel moves on, should it go to a known location or sample somewhere new? Our metrics echo foraging studies where behavior is considered to be exploratory if it alternates between patches and varies over time, and as exploitative if it continues to utilize the same patches and is stable over time[15]. PRT quantifies the dilemma of whether to continue exploiting at the current location or to explore a new location in the hope of increasing the rate of reward[16], often referred to as slow vs. fast exploration[17,18]. In contrast, choice entropy aims to quantify how much effort agents invest in gathering information about their environment, and may be thought of as broad vs. narrow exploration. We make the assumption that greater sampling effort leads to enhanced information about resource distribution. We calculate choice entropy using the information entropy concept[19], which measures the predictability of time series such as human mobility datasets[20], and may be extended to become a measure of exploration[21]. In the context of the bandit-task paradigm, entropy is highest when there are many available portfolio options whose probability of selection is uniform, as the outcome is hardest to predict. Conversely, entropy will be lower when there are fewer portfolio options and/or the probability of selection is skewed in favor of certain options, making the outcome easier to predict[19].

First, we analyze data from a 2-year undisturbed period prior to the closure and we find no relationship, either humped or linear, between performance and exploration. We then analyze data from during the disturbance. We find that the revenues of more-explorative vessels were less adversely impacted by the closures and they were also more likely to continue fishing.

## Results

**Undisturbed period**. The GoM bottom longline fleet was heavily impacted by the closure of prime fishing grounds in 2009 (Fig. 1). We divide our data into two subsets, the undisturbed period prior to the closure and the disturbed period during the closure. Calculating our exploration metrics using the data from the undisturbed period, we find that vessels display a diverse range of individually consistent choice entropy scores (Fig. 2).

We first ask whether a vessel's EETO strategy predicts performance during the undisturbed period. We hypothesize that vessel-level performance should be maximized around intermediate entropy and/or PRT scores, representing strategies that more-evenly balance exploration and exploitation.

Using 2 years of data from the undisturbed period, we measure performance, $P_i$, of each vessel $i$ as its mean revenue across all trips (see "Methods" section, Eq. (2)) after controlling for variations in trip duration and time-varying fleet-wide exogenous variables (Fig. 3a). Fitting a regression model (see "Methods" section, Eq. (5)) with vessel performance modeled as both quadratic and linear functions of entropy and PRT, we find no significant relationship—humped or linear—between either entropy or PRT and performance. Furthermore, model selection

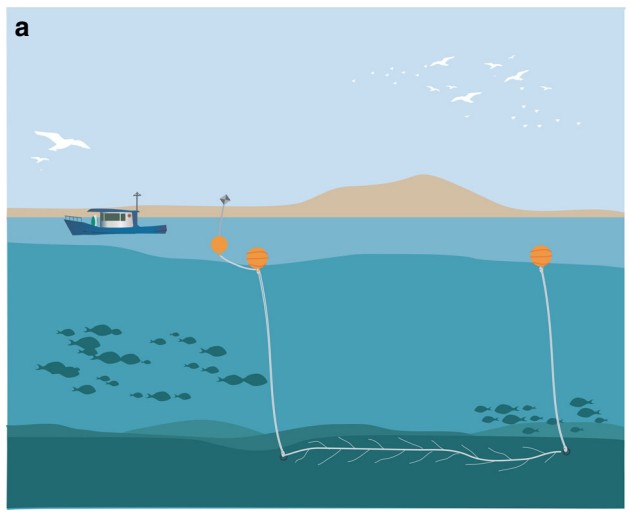

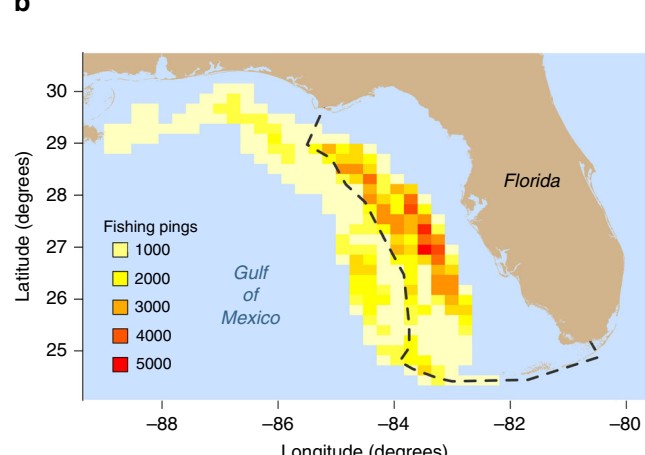

**Fig. 1** Longline fishing in the Gulf of Mexico. **a** Bottom longlines are one of the primary gears used to catch commercially important grouper and tilefish. The longline consists of a line of baited hooks deployed along the seabed in reef areas of the Gulf, which remains in position for a number of hours before being recovered, at which point the fish are removed, the hooks are rebaited and the line is redeployed. **b** Heat map of fishing activity by longline vessels in the eastern Gulf of Mexico during the 2-year period prior to an emergency closure from May to October 2009. Higher fishing activity (darker shading) cells mostly lie within the boundary of the subsequently closed area (dashed line), highlighting the magnitude of the disturbance when many vessels were forced to relinquish their most- utilized fishing grounds. To maintain data confidentiality, locations fished by fewer than three vessels are not shown

using AICs and BICs indicates that all of the entropy and PRT variables can be dropped from the model.

We conclude that more-exploratory vessels performed no better or worse on average during the undisturbed period. The only significant predictors of performance during the undisturbed period were activity ($t_{[103]} = 3.47$; $P < 0.001$), which is a lagged measure to address potential endogeneity issues with performance, and vessel length ($t_{[103]} = 6.24$; $P < 0.001$). See Table 1 for details of covariates and Supplementary Tables 1 and 2, respectively, for results from simplified and fully specified models.

**Disturbed period**. During the disturbance caused by the closure of fishing grounds, we hypothesize that more-exploratory strategies may buffer against adverse impacts, when vessels with diversified portfolios of fishing grounds should benefit from their enhanced knowledge of non-impacted resources.

To test this hypothesis, we first ask whether more-exploratory vessels were more likely to remain in the fishery. Aggregating the choice entropy and PRT scores for all vessels in the fishery before ($N = 106$) and during ($N = 57$) the disturbance, we find that the mean entropy of the fleet significantly increased from 4.1 to 4.6 (Fig. 3b; $F_{[159]} = 10.97$; $P = 0.001$) meaning that higher entropy vessels (broad explorers) were more likely to continue fishing during the disturbance. The mean PRT of the fleet did not change significantly (Fig. 3c; $F_{[159]} = 0.90$; $P = 0.344$) meaning that neither fast nor slow explorers were more likely to continue fishing.

Second, we ask whether exploration has a positive influence on performance during the disturbance. Using the subset of vessels which remained in the fishery during the disturbance, we calculate deviance in performance, $\Delta P_i$, for each vessel $i$ by comparing observed revenues against expected revenues (Eqs. (3) and (4)). Expected revenues are predicted using a business-as-usual scenario, and $\Delta P_i$ values above/below 0 indicate, respectively, better/worse than business-as-usual performance.

To test whether $\Delta P_i$ was positively influenced by entropy or PRT, we fit a linear mixed-effects model (see "Methods" section, Eq. (6)) with vessel identity as a random effect[22] to account for varying numbers of trips among vessels. After controlling for

inter-vessel differences in length, activity, and spatial displacement resulting from the closure, we find that entropy and PRT, respectively, have positive and negative influences on $\Delta P_i$ indicating that vessels with a history of more-exploratory EETO strategies (higher entropy and/or lower PRT) experienced less-adverse impacts on performance during the disturbance. Both entropy and PRT are retained in the model by AIC and BIC selections, although only PRT is significant at $\alpha = 0.05$ (Supplementary Table 1).

Although we find measurable benefits during disturbance to having a history of exploratory behavior, we expect that gains might dissipate over time, which would be consistent with a number of lines of reasoning. For example, economic theory suggests that dissipating advantage may result from exploiters switching strategies to become more exploratory, social network theory suggests that private information may be shared between explorers and exploiters within networks[23] and ecological foraging theory suggests that displaced agents may redistribute themselves based on public information, such as the observed performance of others[24].

To investigate our dissipation hypothesis, we ask whether the buffering effect of higher-entropy strategies diminishes with increasing time from the onset of disturbance. We use a growing-window analysis whereby we iteratively re-fit the $\Delta P_i$ regression model (Eq. (6b), Supplementary Table 1) to subsets of the disturbance data that increase in 10-day increments. We find that choice entropy and displacement are the only variables retained in all models fitted to the five subsets. The benefit of higher-entropy strategies is greatest in the immediate aftermath of the closure (Fig. 3d; $t_{[21]} = 2.841$; $P = 0.01$) and its effect diminishes over time (Fig. 3d). We conclude that higher-entropy vessels (broad explorers) gained the strongest buffering effect in the short term.

## Discussion

There is a rich literature on the EETO in fields ranging from animal foraging[25,26], computer science[1,11], organizational learning[27], neurobiology[3], psychology[2,28], psychiatry[5], and others. Although the concept of explore vs. exploit seems straightforward,

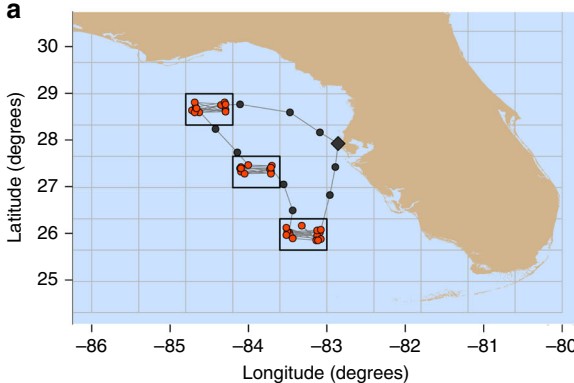

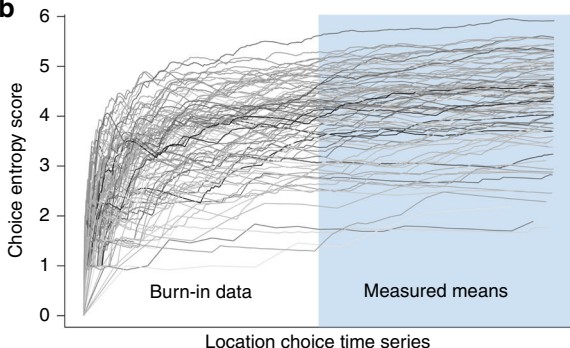

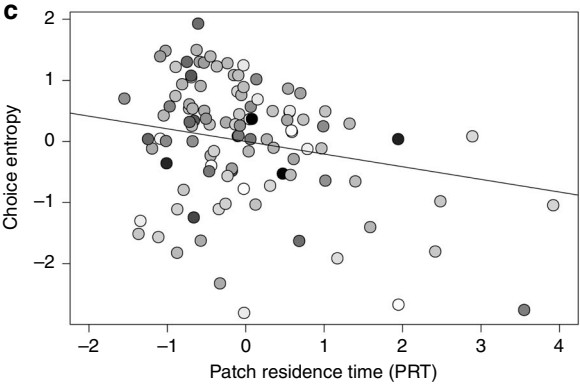

**Fig. 2** Quantifying exploration in vessel movement tracks. **a** The Vessel Monitoring System records hourly positions (pings) of commercial fishing vessels in the Gulf of Mexico. In the simulated vessel track shown, markers represent pings that have been classified into one of three activities: fishing (red circles), transiting (●) or in port (◆) using a supervised learning algorithm trained using data from trips when on-board fisheries observers were present. To discretize fishing grounds, a grid is overlaid on the pings and then cells containing fishing activity are identified, allowing a portfolio of fishing locations to be created for each vessel. The location choices of less-exploratory vessels who repeatedly revisit the same locations are easier to predict (lower choice entropy) than those of more-exploratory vessels who regularly sample new locations (higher choice entropy). **b** Calculated using 2 years of mobility data from before the disturbance, individual choice entropy trajectories approach plateaus for the 106 vessels in our analysis, showing consistent and substantial separation among vessels in exploratory behavior. The first half of each vessel's trajectory is used as a burn-in period, then the mean of the second half is calculated to quantify the vessel's explore-exploit trade-off (EETO) strategy. Vessels with higher choice entropy scores have more-exploratory strategies. **c** There is a weak negative correlation (−0.242; solid line) between choice entropy and patch residence time. Markers show the centred scores for each vessel expressed in standard deviations around zero means. Darker shading indicates higher pre-disturbance performance

there are contrasting approaches to operationalizing the EETO both within and among these fields[15]. Furthermore, there is no universal definition of what constitutes exploratory vs. exploitative behaviors, as this varies not only among fields but is also influenced by measurable factors, such as the spatial scale of analysis, as well as with latent factors such as the internal state of the decision maker[15]. The problem is further complicated by the fact that a given decision may integrate both exploratory and exploitative components. Given these challenges, we place our study in the foraging literature and we use the EETO as an explanatory framework to ask questions of the agents' interactions with the environment rather than to analyze individual choices[15].

Recognizing that it is difficult to classify a given decision as being either more-exploratory or more-exploitative without understanding the internal state of the decision maker, our study concerns the consequences of foraging choices rather than how those choices are made. We use two metrics which capture contrasting but complementary facets of exploration. PRT captures speed of exploration (fast vs. slow explorers), whereas choice entropy captures breadth of exploration (broad vs. narrow explorers). Although all foraging agents will invest in both exploration and exploitation, agents tend to maintain fairly consistent positions along a particular EETO continuum[5]. In any given setting, is not immediately clear where any optimal strategy might be located along an EETO continuum. We hypothesized for the GoM longline fleet that performance might be maximized around intermediate strategies, where the explore–exploit trade-off was more evenly balanced. We found, however, no discernable relationship—either humped or linear—between average payoff and either speed or breadth of exploration during the undisturbed period. This is surprising, given that the motivation for an agent balancing the EETO is generally assumed to be improving long-term performance[5].

Given the characteristics of our setting (changing payoffs and portfolio options), the EETO strategies displayed by our study vessels likely span a range of economically viable options and on average there are no long-run gains from changing strategy. The fact that we do not find an optimal strategy in our natural setting is supported by theoretical investigations, which suggest that performance-maximizing EETO strategies are fundamentally unknowable in systems with highly non-stationary rewards[2,11]. If so, adaptive pressure to develop apparatus capable of optimizing the EETO dilemma would be considerably weakened, although such putative adaptations often motivate optimal foraging hypotheses. However, we cannot completely rule out that our finding is due to a lack of statistical power, although 2494 trips by 106 vessels is a reasonable dataset size.

During the closure, however, we go on to find that both faster and broader explorers experienced less adverse performance deviances, indicating that a history of exploration may deliver benefits during periods of disturbance. Broad explorers were also more likely than narrow explorers to continue fishing, leading to an increase in the mean choice entropy of the fleet (Fig. 3b). Fast explorers, on the other hand, were no more likely than slow explorers to continue fishing during the closure, as indicated by the lack of any significant change in the mean PRT of the fleet (Fig. 3c). The contrasting results from our two metrics highlight the benefits of considering multiple aspects of exploration when investigating the EETO.

A parsimonious interpretation of why broad explorers were more likely to continue fishing is that investment in exploration provides agents with spatially diverse and updated information on resource distribution and dynamics, allowing them to select from a wider pool of alternatives when their preferred grounds are suddenly closed. In the context of the multi-armed-bandit problem, the closure of fishing grounds is analogous to the

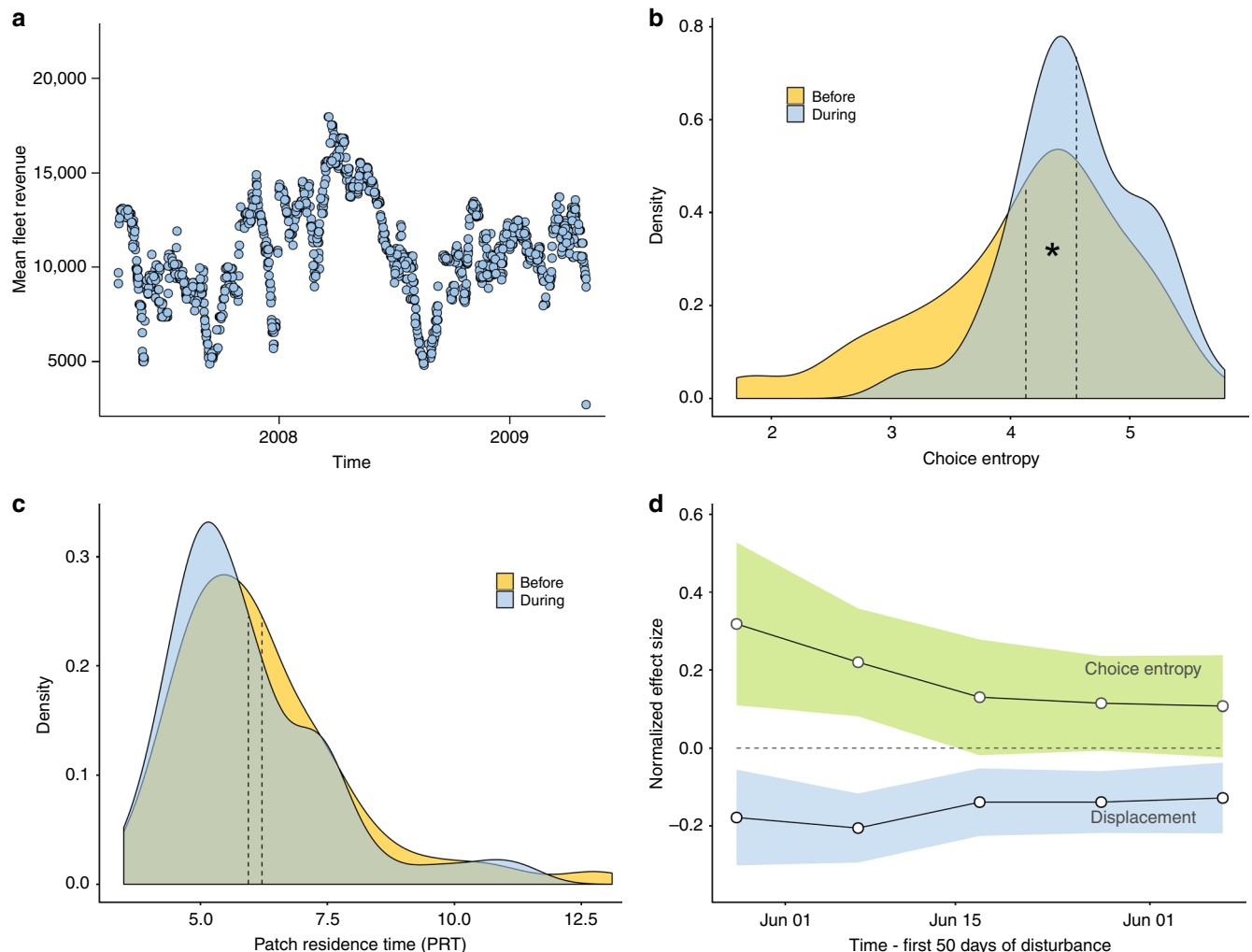

**Fig. 3** Fleet-level revenues and exploration metrics. **a** Temporal variation in fleet performance, reflecting a wide range of environmental and socio-economic factors, which we expect to have similar impacts across the fleet. Longer-period oscillations are likely caused by environmental, ecological, and market-price seasonality, whereas shorter-period oscillations may be caused by weather events. **b**, **c** Densities of entropy scores (**b**) and patch residence time (**c**) for all vessels in the fleet before vs. during the closure (yellow and blue distributions, respectively). Dashed lines show the means of the distributions, and an asterisk indicates when the means are significantly different to each other. Higher entropy vessels (broad explorers) were more likely to remain in the fishery, increasing the average entropy of the fleet during the disturbance. **d** Our regression modeling shows that during the emergency fishery closure in 2009, the most consistent positive predictor of performance deviance was choice entropy (S; green ribbon shows 95% CI). Displacement (blue ribbon), namely the proportion of each vessel's pre-disturbance fishing grounds that were closed, had a negative effect. The effects of both covariates diminish over time. Data shown are from 18 May to 6 July 2009, as there are few records after this period

removal of bandit machines from the portfolio of available options. Although the machine-removal paradigm does not seem to be commonly employed in bandit trials involving human subjects, it would be interesting to know if human EETO mediation is better adapted to buffering such disturbances than to maximizing performance efficiency, given that optimal (or even near-optimal) strategies may be unknowable. In certain fisheries, social networks have been shown to have impacts on fishing behavior[23]. In the context of our study, social networks may diminish the value of personal information held by more-exploratory captains if the information is shared during disturbances, which would erode any putative gains from being more exploratory. We found, however, a significant benefit to investing in exploration irrespective of whether or how information is shared within networks. It is of course possible that social networks are weak within our study system, but other hypotheses are possible. For example, the diminishing benefit observed in more-exploratory vessels may have resulted from the

diffusion of information through social networks. It is also plausible that disturbance changes the value of information within fishing social networks, and information that was openly shared during times of stability may become more guarded. It would be interesting to know whether the subsequent introduction of Individual Fishing Quotas (IFQs) has fundamentally altered the value of social information within the GoM bottom longline fleet, and whether more-exploratory vessels will continue to benefit during future disturbances now that the 'race to fish' has dissipated[29].

Our study contributes to a growing body of work that takes advantage of burgeoning availability to human mobility datasets. In particular, there has been great interest in analyses focused on mapping or modeling aggregated patterns of space-use at the level of populations[20,30–33]. In contrast, our study drills down to consider the perspective of the moving agents themselves, demonstrating that mobility datasets can provide rich long-itudinal records of human decision-making that allow researchers

**Table 1 Covariates used to investigate relationships between exploration and performance prior to and during the spatial closure of Gulf of Mexico longline fishing grounds in 2009**

| Covariate | Description | Interpretation |
|---|---|---|
| $S$ | Choice entropy | Higher values indicate more-exploratory EETO strategies |
| $S^2$ | Choice entropy squared | Quadratic term used to determine if performance is maximized around intermediate EETO strategies that more evenly balance exploration and exploitation |
| activity | Baseline activity level, calculated as the number of longline fishing days for each vessel | Controls for the fact that high-entropy strategies require high levels of fishing activity but high-levels of fishing activity do not necessarily predict entropy, because less-exploratory vessels are often highly active |
| prt | Patch residence time | Higher values indicate less-exploratory EETO strategies, with vessels spending on average longer at each location before moving on |
| $prt^2$ | Patch residence time squared | Quadratic term used to determine if performance is maximized around intermediate EETO strategies that more evenly balance exploration and exploitation |
| length | Vessel length | Controls for the fact that larger vessels have greater mobility potential, allowing them to more easily relocate during the disturbance regardless of entropy |
| displacement | Range displacement, calculated as the proportion of each vessel's pre-disturbance range that was closed to fishing | Controls for the fact that vessels which lost more fishing ground should experience more-negative deviance in performance regardless of their EETO strategy. Higher values indicate greater loss of fishing grounds |

to investigate the underlying processes driving space-use patterns. Although our study asks a question of theoretical interest, agent-level mobility analysis holds considerable promise for applied research. For example, climate change models predict that natural disturbances will increase in frequency and/or severity[34]. Given that disturbance impacts tend to cluster in space, as do socio-economic groups, it is likely that certain groups will be more vulnerable to disturbances. The vulnerability of any group will depend on the non-linear interactions between their mobility capacity and their space use patterns, which themselves depend on access to financial, human, and social capital[35]. Analyzing mobility data at the level of agents could allow scenario planners and policy analysts to disaggregate adaptive capacity within a 'population' and create instruments that explicitly mitigate social inequity arising from impacts on lower income—and thus lower mobility—socio-economic groups.

## Methods

**Datasets**. To test our hypotheses, we use three datasets detailing the behavior of the GoM bottom-longline fleet while targeting commercially important reef fishes. The first dataset consists of more than half a million hourly GPS positions (pings) for 106 vessels, recorded by the Vessel Monitoring System (VMS) that has been implemented fleet-wide since 2006. The second dataset was gathered by independent fisheries observers who accompany a subset of all commercial trips in the GoM, recording the times and locations when fishing gears are deployed and recovered, allowing us to train a machine-learning algorithm to recognize fishing activities. Our third dataset comprises logbook records that detail trip-level data, including revenue earned, which enables us to map behaviors into rewards. We divide the data into two study periods, undisturbed (19 May 2007–18 May 2009) and disturbed (19 May 2009–28 October 2009). GoM bathymetry data were obtained from the National Oceanic and Atmospheric Administration (NOAA). Data analyses were conducted in R (version 3.4.1)[36].

**Processing of logbook data**. All commercial fishing vessels owning a Gulf of Mexico reef fish (GoMRF) permit are required to maintain a logbook recording trip-level data. In the present study, logbook data from 2494 trips by 106 bottom-longline vessels were analyzed. Variables contained in the analysis were vessel ID number, trip duration, gear type, revenue earned, date of landing, species landed, and vessel length. Data were filtered to retain only vessels using bottom-longline gears to target the grouper–tilefish component of the GoMRF fishery. To facilitate performance comparisons among vessels that are conducting fishing trips at different times of the year, it is necessary to control trip-level revenues for temporally variable factors, such as storms and seasonality in market prices. The revenues for each vessel $i$ on each trip $t$ are controlled by including in models the covariate, fleet performance, $F_{-it}$, which is the average revenue of contemporaneous trips by all vessels in the fleet except vessel $i$ during the 2-week period following trip $t$. Two weeks was chosen as a compromise between using too short a window (thus overfitting to limited data) and too long a window (thus failing to capture short-

term events such as storms). Although it would have been preferable to center each trip $j$ symmetrically within its temporal window, doing so would have resulted in miscalculating $F_{-it}$ near the disturbance threshold when data from immediately after the start of the closure would have been incorporated in calculating $F_{-it}$ from trips immediately before the disturbance, and vice versa. Instead, the 2-week window after each trip $j$ was used in calculating $F_{-it}$, and trips from the 2-week window immediately prior to disturbance were deleted. Bias induced by using a skewed window rather than a symmetrical window was minimal, and comparing $F_{-it}$ from the skewed window method (14 days after each trip) to $F_{-it}$ from a symmetrical window method (7 days before and after each trip) gives a Pearson correlation of 0.94 ($t = 184$, df $= 5755$, $P < 0.001$). Activity was quantified as the number of longline fishing days conducted by each vessel, and endogeneity was avoided by using data from the 2-year period, 18 May 2005–17 May 2007, which precedes the study period 18 May 2007–28 October 2009.

**Processing of VMS and observer-program data**. VMS transponders sending hourly or better reports (pings) have been mandated on commercial reef-fish fishing vessels in the GoM since 2006 and were available for all vessels in the grouper–tilefish fishery by early 2007. Each ping consists of the current latitude and longitude of a vessel along with a timestamp, allowing vessel tracks to be mapped with high spatio-temporal resolution. 587,204 pings from 106 longline vessels were assessed for use in the analysis. Pings with GPS coordinates from outside the GoM were deleted. Ping timestamps were converted to POSIX objects with UTC time zone to match VMS data recording protocol. To derive vessel movement speeds, the interval between each ping's timestamp and the preceding timestamp was obtained, the distance between successive pings was calculated using the spherical geometry function, distRhumb, in the R-package, geosphere[37], and then speed was expressed as a linear distance over time (m s$^{-1}$). Derived vessel speeds above an arbitrary threshold of 20 m s$^{-1}$ were assumed to result from errors and were deleted. Depth at each ping location was extracted from the NOAA ETOPO1 database using the R-package marmap[38].

**Identification of fishing grounds**. We use supervised learning to identify fishing grounds by training a random forest ensemble[39,40] to discriminate fishing activity from other behaviors (transiting, moored, etc.) in vessel movement tracks. To convert the continuous space of the tracks into discrete fishing grounds, we rasterize the fishing pings using a $30 \times 30$ grid fitted to the extent of the entire fleet dataset and then create a portfolio of fishing location choices for each vessel by identifying the raster cells within which it had fished during the undisturbed period (Fig. 2a).

We create a supervised-learning dataset using the subset of bottom-longline fishing trips for which on-board observer data were available, providing a record of VMS points where vessels were known to be fishing. We split this dataset evenly into 'training' and 'testing' subsets. We build a random forest classifier the R-package randomForest[40] and fit it to the 'training' subset, whereby a suite of vessel movement variables (speed, turning angle, depth, time of day, etc.) is passed to the classifier, so that it learns to identify the characteristics of vessels engaged in fishing. We then use the reserved 'testing' subset of the observer data to cross-validate the performance of the classifier, which achieves >90% balanced accuracy, namely the mean of the true positive and true negative rates, whereby the accuracy score is penalized for incorrect labeling as well as rewarded for correct labeling. We use the validated random forest to classify the entire VMS dataset into locations

representing either fishing or non-fishing behavior. A detailed description of our supervised-learning training and testing protocol is available in the literature[41].

**Quantification of EETO strategy**. We use two metrics to quantify EETO strategy, namely PRT and choice entropy. To develop the PRT score for each vessel, we use its portfolio of pre-disturbance fishing locations (grid squares) identified by the random forest classifier. We calculate the mean time interval that each vessel remains within each fishing location before moving to a subsequent fishing location or returning to port. We quantify choice entropy using the portfolio of fishing locations for each vessel (Eq. (1)). The metric can change through time as the number and frequency of fishing locations of the vessels change. From each vessel's choice entropy trajectory, we calculate the mean value, $S$, after a burn-in period (Fig. 2b). Vessels with higher $S$ scores have more-exploratory EETO strategies, maintaining more-diverse portfolios of fishing locations, many of which they regularly resample. We use the portfolio of fishing locations for each vessel, $i$ to create a chronologically ordered visitation time series, $L_i$. We iteratively extract incrementally increasing subsequences of $L_i$ and calculate the entropy of each subsequence. For example, the tenth subsequence of $L_i$ comprises the first 10 location choices by vessel $i$, and the entropy of the subsequence increases with the number of unique locations in $L_i$ and thus decreases if certain locations are visited more often than others. The choice entropy of vessel $i$ at location $m$ is calculated using the subsequence of location choices, $L_{il=1}^m$, such that:

$$S_{im} \equiv -\sum_{j=1}^{N_{im}} f_i(j) \log_2 f_i(j) \qquad (1)$$

where $N_{im}$ is the number of unique locations fished at by vessel $i$ until the $m$th element of the time series, and $f_i(j)$ is the frequency with which vessel $i$ fished at location $j$ in the subsequence. Iteratively calculating $S_{im}$ for the entire location time series, $L_i$, yields a trajectory of choice entropy scores (Fig. 2b). From each vessel's choice entropy trajectory, we calculate the mean entropy score, $S$, after a burn-in period (Fig. 2b), which we use as our choice entropy metric.

The intuition of using entropy to quantify fisheries exploration is that it captures the non-stationary nature of fish stocks which fluctuate both in space (across habitats) and time (across seasons). Vessels with the highest entropy will be those that maintain a large portfolio of fishing locations (spatial exploration) which they regularly resample (temporal exploration), providing them with a spatially diverse and up-to-date pool of information on resource distribution and dynamics. Conversely, vessels with the lowest entropy will be those that exploit a small portfolio of locations and/or skew their fishing effort towards a small subset of their overall portfolio, handicapping them with spatially limited and/or out-of-date resource information.

Although all vessels will invest time both in exploring and exploiting, some demonstrate consistently more-exploratory strategies than others, as is clear from the vertical separation of the entropy trajectories (Fig. 2b). The correlation between PRT and choice entropy is weak (Pearson coefficient, −0.242; $P < 0.001$; Fig. 2c) indicating that these metrics capture different facets of exploratory behavior. The total fleet/size of longline vessels active during our study period was 125, 19 of which were dropped for having sparse and erratic data (Supplementary Fig. 1), leaving 106 vessels.

**Sensitivity of metrics to grid (raster) size**. To test the sensitivity of the results to the selected grid size ($30 \times 30 = 900$ cells), choice entropy was recalculated using grids ranging from $20 \times 20 = 400$ to $95 \times 95 = 9025$ cells, which represents more than an order of magnitude difference in scale. Pairwise Spearman correlation tests were then performed between the variable scores for each vessel resulting from our chosen grid size and each of the other grid sizes (Supplementary Fig. 2).

**Quantifying displacement**. Not all vessels use the same fishing grounds and therefore not all vessels will have been equally affected by the closure. Among-vessel differences in loss of fishing grounds were controlled for using a variable, displacement, which quantifies the proportion of fishing grounds lost by each vessel during the disturbance, using spatial objects created and analyzed with the R-package, sp.[42] A spatial polygon object was created from the vertices of the closure boundary[14] and a spatial-points object was created from the fishing locations for each vessel during the 2-year pre-disturbance period. The number of pre-disturbance fishing locations that fell within the boundary of the subsequently closed area was quantified and expressed as a proportion of the total number of fishing locations during the pre-disturbance period. For instance, a vessel scoring a displacement value of 0.8 would have lost 80% of its pre-disturbance fishing locations during the closure, and would be expected to be more adversely affected than a vessel which lost only 5% of its pre-disturbance locations.

**Modeling framework**. A straightforward way of modeling a system before vs. during a disturbance would be to include in the model a binary 'dummy' variable, which indicates the period to which each record belongs. However, the nature of our data necessitated a more complex multistage process, as most vessels have few records—often only a single record—during the disturbed period and thus there is little or no within-vessel replication. We handle this by first modeling fishing

payoff (performance) for each vessel prior to the disturbance, then using this model in a business-as-usual scenario to predict what performance should have been during the disturbance if the vessel were unaffected, and finally comparing these predictions against the observed performance for each vessel. A conceptual diagram of our statistical modeling framework is shown in Supplementary Fig. 3. Details and equations are as follows:

**Quantifying undisturbed performance**. We model the performance of vessel $i$ on each trip $t$ during the undisturbed period using revenue observations ($Ru_{it}$) to fit a linear model with vessel identity, $V_i$, as a fixed effect such that:

$$Ru_{it} = P_i V_i + \beta_1 D_{it} + \beta_2 F_{-it} + \varepsilon_{it} \qquad (2)$$

The resulting estimate of $P_i$ is the mean performance of vessel $i$ across all trips while controlling for trip duration, $D_{it}$, and contemporaneous fleet performance, $F_{-it}$ (where $-i$ reflects that the measure explicitly omits vessel $i$ from the calculation) and is our measure of undisturbed performance. When fitting linear models in R, the fixed effect estimates are calculated relative to a reference vessel, which is the model intercept. We correct this by calculating the absolute performance for each vessel $i$ by adding its estimate, $P_i$, to the model intercept.

**Quantifying disturbed performance**. We first control for $D_t$ and $F_t$ in the raw revenue values, giving measured performance on disturbed trip $t$, such that:

$$Md_{it} = Rd_{it} - \hat{\beta}_1 D_{it} - \hat{\beta}_2 F_{-it} \qquad (3)$$

where $Rd_{it}$ is the disturbed revenue for vessel $i$ on trip $t$, and $\hat{\beta}_1$ and $\hat{\beta}_2$ are coefficients estimated from Eq. (3). We then predict the expected performance for vessel $i$ on disturbed trip $t$, $Ed_{it}$, by re-parameterizing Eq. (2) with the trip-level values for $D_{it}$ and $F_{-it}$ using the native R-function, predict. Finally, we standardize the deviance in performance for vessel $i$ on trip $t$ by dividing the measured performance by the expected performance, then rescaling the result so deviances are centered on zero such that:

$$\Delta P_{it} = \frac{Md_{it}}{Ed_{it}} - 1 \qquad (4)$$

Thus positive/negative values of $\Delta Pt_{it}$ indicate better/worse than business-as-usual performance. When aggregated at fleet level to investigate whether greater exploration predicts less-negative deviances in performance during the disturbance, the $\Delta P_{it}$ scores approximate Gaussian distributions (Supplementary Fig. 4). Because some vessels completed more than one trip during the disturbance, a linear mixed effects model is used with vessel identity declared as a random effect to avoid pseudo-replication from non-independent $\Delta P_{it}$ scores.

**Model fitting**. Each continuous covariate was normalized by rescaling units as standard deviations around its own mean using the native $R$-function, scale, and data were explored following the protocol of Zuur et al. [22]. Outliers were checked using Cleveland dot-plots, and a single revenue outlier was removed for being ~8 times greater than the second-largest value. Homogeneity of variance was checked using boxplots, and violations of homogeneity in the raw revenue data were corrected when fitting Eq. (2) by specifying log-link functions and checking plots of model residuals vs. fitted values for lack of structure. Error distribution was checked for approximate normality using $qq$ plots. The number of trips varied among vessels in the disturbance-period data, and the resulting pseudo-replication was handled by fitting a mixed-effects model[22] with vessel identity as a random effect. Stepwise model selection was undertaken, whereby model terms were sequentially dropped and each simpler model was compared against the more complicated model using AICs and BICs. Simpler models that did not result in increased AIC or BIC values were selected and the process was repeated.

We investigate whether pre-disturbance performance was predicted by EETO strategy by fitting the linear regression model,

$$P_i = \alpha + \beta_1 S_i + \beta_2 S_i^2 + \beta_3 prt_i + \beta_4 prt_i^2 + \beta_5 activity_i + \beta_6 length_i + \varepsilon_i \qquad (5)$$

where $S_i$ and $S_i^2$ test, respectively, for linear or quadratic relationships between entropy score and performance $P_i$; $prt_i$ and $prt_i^2$ test, respectively, for linear or quadratic relationships between PRT and performance. The quadratic terms test for humped relationship where performance is maximized around intermediate strategies that more evenly balance exploration and exploitation.

During the disturbance event, we investigate whether a history of exploration (higher choice entropy or lower PRT) had a positive influence on performance deviance, $\Delta P_i$, while controlling for activity, vessel length and spatial displacement resulting from the closure. $\Delta P_i$ values above/below 0 indicate, respectively, better/ worse than business-as-usual performance (Supplementary Fig. 3). We use a mixed-effects model to avoid pseudo-replication caused by varying numbers of

trips among vessels:

$$\Delta P_i = (\beta_0 + b_{0i}) + \beta_1 S_i + \beta_2 S_i^2 + \beta_3 \text{prt}_i + \beta_4 \text{prt}_i^2$$
$$+ \beta_5 \text{activity}_i + \beta_6 \text{length}_i + \beta_7 \text{displacement}_i \tag{6}$$
$$+ \varepsilon_i; \text{ Random intercept, } b_{0i} \sim \text{Normal}(0, \sigma_i^2)$$

**Growing window analysis.** To investigate temporal changes in the variables predicting deviance in performance, $\Delta P_i$, during the disturbance, a growing (or expanding) window approach was used. A fully specified model (Eq. (6b), Supplementary Table 1) was fitted to incrementally increasing subsets of the disturbance period data, with 10 days of additional data being added at each iteration. 10-day increments were chosen as a compromise between obscuring short-term changes in effect sizes by integrating data over a longer period, and overfitting by including too few data in the first few iterations. Model results were plotted using the R-package, ggplot2[43], which was also used to plot changes in fleet level entropy. Models were fitted to the first 50 days of the disturbance only as data became sparse after this time (Supplementary Fig. 5). Density plots of the response variable, performance deviance, $\Delta P_i$, are shown in Supplementary Fig. 4, demonstrating that the variable follows an approximately Gaussian distribution without being piled against the bounds.

**Entropy score is not confounded by fishing depth.** Although the GoM reef fish longline fishery consists of vessels targeting shallow-water fish, or deep-water fish, or a combination of the two, and deeper waters are systematically further offshore in the GoM, there is no systematic relationship between fishing depth and our entropy metric (Supplementary Fig. 6).

**More-exploratory vessels incur higher travel costs.** Investment of resources in exploration introduces risks in terms of expected payoffs and likely requires additional travel costs relative to an exploitation strategy. To investigate whether vessels with higher choice-entropy EETO strategies incurred higher costs relative to revenues, we regress entropy score, $S$, on revenue per kilometer traveled. We find that higher entropy vessels traveled farther per unit of revenue ($F_{[104]} = 17.29$, $P < 0.001$), confirming that there are measurable costs to investment in exploration (Supplementary Fig. 7).

## Data availability
The study data were obtained under a contractual agreement with the U.S. National Marine Fisheries Service (NMFS). The agreement prevents distribution of personally identifiable information, including variables directly included in the analysis. These data are archived at NOAA's Southeast Fisheries Science Center. Researchers under a contractual agreement with NMFS can access the data provided a nondisclosure agreement is signed.

## Code availability
The full R-code used to conduct the analysis is available at GitHub.

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

## Acknowledgements

We thank Iliana Chollett for production of Fig. 1a as well as for providing comments during study development, as did Marcy Cockrell, Jordan Watson, The NatuRE Policy Lab and The Andy Sih Lab at UC Davis. We are grateful to Elizabeth Scott-Denton at the SEFSC Galveston Lab for providing the observer data. We also thank Peter M. Allen, David L. Barack and one anonymous reviewer for improving the manuscript during peer review. Figure 1a uses elements from the Integration and Application Network[44]. The study was funded by NSF Coastal SEES Grant #1325452 (S.A.M. and J.N.S.); National Academy of Sciences, Gulf Research Program Data Synthesis Grant #2000007631 (S.A.M. and J.N.S.); Spatial Economics Toolbox for Fisheries (FishSET) Project (A.H.).

## Author contributions

S.O., J.N.S., O.S., and S.A.M. designed the study; S.O., J.N.S and O.S. analyzed the data; M.D., A.H., A.S., and L.P. provided data products and developed the study concept; all authors wrote the manuscript. S.O., J.N.S and O.S. contributed equally to the study. The scientific results and conclusions, as well as any views or opinions expressed herein, are those of the authors and do not necessarily reflect the views of NOAA or the United States Department of Commerce.

## Additional information

**Competing interests:** The authors declare no competing interests.

