## [Peer Review File · Nature Communications]

Reviewers' comments:

Reviewer #1 (Remarks to the Author):

This paper utilizes a fishing data set in the Gulf of Mexico to assess the impact of changes in the environment on fishing revenue. The authors investigate whether differences in fishing strategy affect this revenue before and during a disturbance event where typical fishing grounds were off limits. They conclude that fishing strategy—whether ships are more or less exploratory—had little influence before disturbance but strongly impacted revenue during the disturbance, with more exploratory vessels faring better than less exploratory ones.

Overall score: 7/10.

Recommend: Moderate R&R.

Overall comments: this paper features a unique data set and asks interesting questions. There is no question in my mind that this paper should be published following suitable re-analyses and revisions. Their conclusion may change as a result of the recommendations, but that's a good thing, if so! The recommended changes are mostly with regard to analysis, although there is one major conceptual issue. However, this is a wonderful opportunity to explore real world foraging decisions in a modern setting. The article will appeal to scientists of diverse backgrounds including economics, neuroscience, animal behavior, neuroeconomics, ecology, climate scientists, philosophy (namely the impact of adaptations from our evolutionary past on current decision making and on group decision making), and more. The journal would be remiss to have the paper shuttled to another, lesser journal.

Specific comments:

Note: some of these comments are suggestions, others are questions, and some recommend confirmatory or revised analyses.

1. There is a rich foraging literature on influence of changes in the environment on foraging strategy in birds. See, e.g., the role of temperature in foraging in birds, such as Bateson 2002 or Caraco et al. 1990 in juncos. The Caraco article in particular is a classic.
2. The authors utilize revenue as the economic variable of interest. But what about costs? What about costs? The CBA will be expected payoffs minus expected costs. For example, 'more exploratory' ships in their study may amount to more efficient ships whereby the cost of moving is much less than other ships. Unless 'revenue' in their analysis already takes into costs?
3. Grid size may affect their results. The authors utilized a 30x30 grid. Now, the authors did vary the size of this grid by +/- 10%. This is good, but waaaaay more exploration of how the grid size

changes the results would be advised. So I would test 3 orders of magnitude (e.g., 3x3, 30x30, 300x300, although 3x3 makes very little sense, so maybe 30x30, 300x300, and 3000x3000). This will take a long time to analyze, but the methods should be the same, so it is just computer time. The concern is that with the wrong size grid, what looks like exploitative strategy will in fact be exploratory.

In addition, I wonder about a different sort of analysis. Is there a continuous variable that could be used, such as total distance traveled? You could just integrate over path length and see if that is influenced by their covariates. Discretizing is fine; the concern is that they overlook other behaviors that truly reflect exploration. For example, in rats, exploration often results in more turning. Do the boats turn more? Questions like that.

4. Major Conceptual Issue. While the description of what their entropy measure assesses is accurate, this does not neatly map on to EE. For example, in RL, exploratory choices are defined as non-exploitative ones, and exploitative choices are defined as choosing the action with the highest discounted expected return in the current state. Conceivably, a boat can be in a succession of states, each of which demands a different location choice. That might appear exploratory but, in fact, is exploitative.

One way to quantify: Kalman filters (see, e.g., Daw et al. 2006; Pearson et al. 2009). Another way to quantify: simulate 10K RL agents and let them roam the virtual seas, see how their simulated catch goes, and assign a value to each location. To get the values for the catches, look at a boat that actually fished there on that day. A third way to quantify: first principles: use the Bellman equations with location values computed from the actual historical record. Note that some way of defining a state will be necessary.

For a journal like Nature Comms, all three should be attempted, if possible.

Coming up with a motivated formal definition for exploration is absolutely essential for asking EE questions. The authors assume that variability is exploratory. That is simply not always the case. Without such a more formal approach, assessing the merits of the paper are difficult.

There is an alternative; the authors could simply say they are operationalizing exploration using their entropy measure. But this would lead to some general questions about the relation between this work and the conceptual approach embodied by the RL literature on EE.

5. Please change 'explorative' to 'exploratory'.

6. The authors might try to develop an agent-based foraging model. How autocorrelated are the fish schools? If they are autocorrelated, then the environment should be 'patchy'—featuring patches or clumps of resources in the environment. The decisions to move on or not might reflect the current catch intake rate and distances/times to the next patch. There is a major motivation to analyse EE decisions as foraging. See, e.g., Charnov 1976.

7. Pleased to note that the authors looked at the residuals for structure!

8. Regression notes:

- a. I would include all the covariates in a single regression. Perhaps that was what in fact was done?
- b. I would include weather, market price of fish, gasoline prices, and so forth in the regression as well.
- c. Please use BIC as well as AIC to confirm best fit models.
- d. More generally, there are a million covariates to control for. Run the kitchen-sink model. It sucks because it will eat up tons of your variance, but it is the right thing to do if you are going to use these models to infer effects.

If the kitchen-sink model does not work, then you say that, and start doing fancier things—running power analyses, choosing more complex link functions, and the like.

9. Is there really no work on removing bandits? Steyvers et al. have extensively studied different bandit problems—the authors should confirm this is the case.
10. Pleased to see that the data were curated! Good data is curated. However, please note in the text how many boats were excluded (and if there is room, why).
11. The supplemental plots and main plots need more labelling (blue = X, green = Y, descriptions of the axes, etc.).

Signed: David L Barack, Columbia University

Reviewer #2 (Remarks to the Author):

This paper reflects the improved data availability for people studying fisheries. The behaviour of individual boats can be followed as they go out on fishing trips. The study concerns long-line fisheries and seems to be both interesting and correct. My own studies concerned the groundfish fisheries of Nova Scotia. For these we developed a model of the fisheries which generated the movement of fleets and their resulting catches and profits. Interesting differences emerge between the fishing studied in this paper and the one we studied in Nova Scotia. We certainly raised the importance of explore/exploit behaviour. The main difference seems to be concerning the importance of 'social' factors. In Nova Scotia, fishing fleets were important rather than individual boats. Fishing affected the stocks quite rapidly and information of the spatial location of fish stocks was kept within fleets where possible. Coded messages and misleading information were used to stop knowledge spreading to rival fleets. Fleets could have a statistical distribution of responses varying from completely random to all boats going to the best place they know of. Fleets could be

composite – with some explorers and some exploiters. In case anyone was interested, publications were 1987 Allen P M and McGlade J M , “Modelling Complex Human Systems: A Fisheries Example”, European Journal of Operational Research 30 (1987), pp 147-167.

I think that the paper is fine and offers an interesting analysis of this long line fishery, and the changing pay-off for exploration and exploitation. Data availability for individual boats opens up opportunities for new knowledge. The work and ideas described seems very clear and the conclusions seem correct. It should be published. Peter M Allen

Reviewer #3 (Remarks to the Author):

The manuscript assesses real-world payoffs of contrasting explore/exploit trade-off (EETO) strategies using longline fishing in the Gulf of Mexico, leveraging a unique natural experiment where a large portion of prime fishing grounds was closed for five months in 2009 (due to excessive bycatches of endangered species of sea turtle) and many affected vessels were forced to relinquish habitually fished grounds and either fish elsewhere or retire during the closure. The authors assume that strategies investing too heavily towards either end of the spectrum are likely to be suboptimal and that before the disturbance, vessel-level performance may be maximized around intermediate strategies that more-evenly balance exploration and exploitation; however, during the disturbance, the more-explorative strategies should buffer against adverse impacts, when vessels with diversified portfolios of fishing grounds should benefit from their enhanced knowledge of non-impacted resources. The authors quantify EETO strategy using information entropy and develop cumulative entropy metric using the portfolio of fishing locations for each vessel. The findings show that in stochastic natural systems characterized by non-stationary rewards, the role of exploration in buffering against disturbance may be greater than its assumed role in improving performance.

This is a well-designed study with high quality data. The interpretation of the data is also well justified and the conclusions are novel and well supported by the results, and will be of interest to a wide readership. I recommend acceptance of the manuscript in Nature Communications essentially as is.

I only have a few minor comments:

- Is 106 the total longline fleet in the US GoM or a sample? Perhaps a short description of fishery would be useful in order to calibrate the representativeness of the results.

- During the closure of the fishery, did the boats receive any subsidy or were they compensated in any way? If so, could this affect the disturbance level and strategy of the boats?

Reviewers' comments

Reviewer #1 (Remarks to the Author):

This paper utilizes a fishing data set in the Gulf of Mexico to assess the impact of changes in the environment on fishing revenue. The authors investigate whether differences in fishing strategy affect this revenue before and during a disturbance event where typical fishing grounds were off limits. They conclude that fishing strategy—whether ships are more or less exploratory—had little influence before disturbance but strongly impacted revenue during the disturbance, with more exploratory vessels faring better than less exploratory ones.

Overall score: 7/10.

Recommend: Moderate R&R.

Overall comments: this paper features a unique data set and asks interesting questions. There is no question in my mind that this paper should be published following suitable re-analyses and revisions. Their conclusion may change as a result of the recommendations, but that's a good thing, if so! The recommended changes are mostly with regard to analysis, although there is one major conceptual issue. However, this is a wonderful opportunity to explore real world foraging decisions in a modern setting. The article will appeal to scientists of diverse backgrounds including economics, neuroscience, animal behavior, neuroeconomics, ecology, climate scientists, philosophy (namely the impact of adaptations from our evolutionary past on current decision making and on group decision making), and more. The journal would be remiss to have the paper shuttled to another, lesser journal.

Dr. Barack, thank you for your kind words and your thorough review. We hope we have addressed your comments satisfactorily.

Specific comments:

Note: some of these comments are suggestions, others are questions, and some recommend confirmatory or revised analyses.

Note: To aid in the review process, we have changed the original order of the comments, bringing together #1, #4 and #6 as they consider a common issue and may be addressed simultaneously. Comment numbering remains unchanged.

1. There is a rich foraging literature on influence of changes in the environment on foraging strategy in birds. See, e.g., the role of temperature in foraging in birds, such as Bateson 2002 or Caraco et al. 1990 in juncos. The Caraco article in particular is a classic. Please see our reply that follows Comment 6.

4. Major Conceptual Issue. While the description of what their entropy measure assesses is accurate, this does not neatly map on to EE. For example, in RL, exploratory choices are defined as non-exploitative ones, and exploitative choices are defined as choosing the action with the highest discounted expected return in the current state. Conceivably, a boat can be in a succession of states, each of which demands a different location choice. That might appear exploratory but, in fact, is exploitative. One way to quantify: Kalman filters (see, e.g., Daw et al. 2006; Pearson et al. 2009). Another way to quantify: simulate 10K RL agents and let them roam the virtual seas, see how their simulated catch goes, and assign a value to each location. To get the values for the catches, look at a boat that actually fished there on that day. A third way to quantify: first principles: use the Bellman equations with location values computed from the actual historical record. Note that some way of defining a state will be necessary. For a journal like Nature Comms, all three should be attempted, if possible. Coming up with a motivated formal definition for exploration is absolutely essential for asking EE questions. The authors assume that variability is exploratory. That is simply not always the case. Without such a more formal approach, assessing the merits of the paper are difficult. There is an alternative; the authors could simply say they are operationalizing exploration using their entropy measure. But this would lead to some general questions about the relation between this work and the conceptual approach embodied by the RL literature on EE. Please see our reply that follows Comment 6

6. The authors might try to develop an agent-based foraging model. How autocorrelated are the fish schools? If they are autocorrelated, then the environment should be 'patchy'—featuring patches or clumps of resources in the environment. The decisions to move on or not might reflect the current catch intake rate and distances/times to the next patch. There is a major motivation to analyse EE decisions as foraging. See, e.g., Charnov 1976.

These are great points. Given the breadth of concepts and contrasting definitions of EE behavior across multiple disciplines ranging from organizational learning to microbiology, explicitly placing our work in the literature is good advice. Furthermore, we agree that the foraging literature (comment #6) is a better place for our study than RL (comment #4), as the two fields are not well reconciled in EE concepts. To explicitly place our study in the foraging literature, we add additional analyses and text that allows our work to link more closely to Charnov's Marginal Value Theorem (comment #6). Our revised analysis incorporates patch residence time (PRT) for each vessel, which quantifies how long each vessel spends on average at a given location before moving on to the next. As with our entropy variable, we test for both linear and quadratic relationships between PRT and performance. Adding PRT to our analysis allows us to capture an additional dimension of EE behavior, namely whether some agents

consistently switch from exploiting to exploring faster than others. In addition to calculating PRT for each vessel and refitting our statistical models with the new variable while using both AIC and BIC comparisons, we have added corresponding text to the Methods and Results. The revised results table (S1) is below and the fully specified models have been included in a new table, S2.

Table S1. Results from model fitting. Equations are fitted by stepwise selection using AICs and BICs, using centered covariates (i.e. expressed in standard deviations around zero means). The change in AIC/BIC is reported for dropped covariates, which are listed in order of deletion. Coefficient estimates are provided for covariates that are retained in optimal models. See Table 1 for details of covariates. Equation numbers as per main text.

Eqn. 3. $P_i = \alpha + \beta_1 S_{\mu i} + \beta_2 S_{\mu i}^2 + \beta_3 prt_i + \beta_4 prt_i^2 + \beta_5 activity_i + \beta_6 length_i + \varepsilon_i$

	Effect size	Standard error	t-value	P	ΔAIC	ΔBIC
prt	-	-	-	-	-1.99	-4.66
entropy, S_μ	-	-	-	-	-1.99	-4.66
S_μ²	-	-	-	-	-0.82	-3.49
prt²	-	-	-	-	-0.96	-3.62
intercept	10905	366.9	29.722	< 0.001	-	-
activity	1281.6	368.5	3.47	< 0.001	-	-
length	2302.1	368.7	6.244	< 0.001	-	-
Degrees of freedom = 103						

Eqn. 4. Mixed effects model fitted to first 50 days of disturbance

$$\Delta P_i = (\beta_0 + b_{0i}) + \beta_1 S_{\mu i} + \beta_2 S_{\mu i}^2 + \beta_3 prt_i + \beta_4 prt_i^2 + \beta_5 activity_i + \beta_6 length_i + \beta_7 displacement_i + \varepsilon_i$$

Random intercept, $b_{0i} \sim \text{Normal}(0, \sigma_i^2)$ (vessel ID as random factor)

	Effect size	Standard error	t-value	P	ΔAIC	ΔBIC
prt²	-	-	-	-	-1.91	-4.5

S_μ^2	-	-	-	-	-1.74	-4.32
activity	-	-	-	-	-1.8	-4.38
intercept	-0.343	0.053	-6.543	< 0.001	-	-
entropy, S_μ	0.107	0.067	1.603	0.115*	-	-
length	0.056	0.041	1.387	0.171*	-	-
displacement	-0.128	0.046	-2.767	0.008	-	-
prt	-0.103	0.049	-2.062	0.044	-	-

Degrees of freedom = 50

Eqn. 4b. As eqn. 4, but fitted to data subset from first 10 days of closure.

	Effect size	Standard error	t-value	P-value	δAIC	δBIC
S_μ^2	-	-	-	-	-1.84	-3.06
prt^2	-	-	-	-	-0.33	-1.55
length	-	-	-	-	-0.38	-1.59
intercept	-0.576	0.087	-6.591	< 0.001	-	-
entropy, S_μ	0.318	0.112	2.841	0.01	-	-
displacement	-0.178	0.066	-2.708	0.014	-	-
activity	0.134	0.062	2.168	0.042	-	-
prt	-0.125	0.092	-1.363	0.188*	-	-

Degrees of freedom = 20

*AIC and BIC both support retaining in the model.

We have also added the following text to the Introduction:

“Recognizing that it is difficult to classify a given decision as being either more-exploratory or more-exploitative without understanding the internal state of the decision maker, our study concerns the consequences of foraging choices rather than how those choices are made. To

capture contrasting aspects of exploratory behavior, we use the time series to calculate two metrics: patch residence time (PRT) and choice entropy. These metrics focus respectively on each of two nested fishing decisions: “Do I continue fishing at my current location or move on?” and “If I move on, do I go to a known location or do I sample somewhere new?” Our metrics echo foraging studies where “Behavior is interpreted as exploration if it alternates between patches or options, is unfocused, and is variable over time, [or] as exploitation if it remains within a patch or option, is focused, and is stable over time.” (Mehlhorn et al 2015). PRT relates to analyses of whether to continue exploiting at the current location or to explore a new location in the hope of increasing the rate of reward (Charnov 1976). PRT is calculated as the mean duration that each vessel remains within a given patch before moving on, as we expect that some agents may systematically remain longer than others at a given location. In contrast, choice entropy aims to quantify how much effort agents invest in gathering information about their environment and allows us to ask whether that information provides measurable advantages in stable vs disturbed conditions. We expect that some agents may explore locations across space and time more widely (higher choice entropy) than other agents, and we make the assumption that greater sampling effort leads to enhanced information about resource distribution.”

We go on to clarify the conceptual challenges identified by the Reviewer by expanding the Discussion to include the following opening paragraph:

“There is a rich literature on the EETO in fields ranging from animal foraging (Caraco et al 1990; Bateson 2002), computer science (Gittins 1974; Whittle 1988), organizational learning (March 1991), neurobiology (Barack and Gold 2016), psychology (Cohen et al 2007; Wilson et al 2014), psychiatry (Addicott et al 2007) and others. Although the concept of explore vs exploit may seem straightforward, there are contrasting approaches to operationalizing the EETO both within and among these fields (Mehlhorn et al. 2015). Furthermore, there is no universal definition of what constitutes exploratory vs exploitative behaviors, as this varies not only among fields but is also influenced by measurable factors such as the spatial scale of analysis as well as with latent factors such as the internal state of the decision maker (Mehlhorn et al. 2015). The problem is further complicated by the fact that a given decision may integrate both exploratory and exploitative components. Recognizing these challenges, we place our study in the foraging literature and we use the EETO as an explanatory framework to ask questions of the agents’

interactions with the environment rather than to analyze individual choices (Mehlhorn et al 2015).”

In placing our study more firmly within the literature, we have added the following new references:

- Bateson, Melissa (2002) "Recent advances in our understanding of risk-sensitive foraging preferences." *Proceedings of the Nutrition Society* 61.4: 509-516.
- Caraco, Thomas, et al. (1990) "Risk-sensitivity: ambient temperature affects foraging choice." *Animal Behaviour* 39.2: 338-345.
- Charnov, Eric L. (1976) "Optimal foraging, the marginal value theorem." *Theoretical Population Biology* 9(2): 129-136.
- March, James G (1991) "Exploration and exploitation in organizational learning." *Organization Science* 2.1: 71-87.
- Mehlhorn, Katja, et al. (2015) "Unpacking the exploration–exploitation tradeoff: A synthesis of human and animal literatures." *Decision* 2.3: 191.
- Wilson, Robert C., et al. (2014) "Humans use directed and random exploration to solve the explore–exploit dilemma." *Journal of Experimental Psychology: General* 143.6: 2074.

To reply specifically to Comment #6: Although building an ABM as suggested is an interesting idea, we feel that it is beyond the scope of this paper and would be unlikely to alter the present conclusions.

2. The authors utilize revenue as the economic variable of interest. But what about costs? The CBA will be expected payoffs minus expected costs. For example, 'more exploratory' ships in their study may amount to more efficient ships whereby the cost of moving is much less than other ships. Unless 'revenue' in their analysis already takes into costs?

We agree that agents are weighing costs and benefits whenever they choose locations. Given that most of the vessels in the fleet utilize similar fishing gears and technology, it is unlikely that there will be differences in the cost of moving a vessel from one location to the next, everything else being equal (distance, currents, winds etc.). The expected benefits associated with a given site, however, could differ due to differences in catching ability (e.g., quality of the fishing skipper). In terms of total travel costs on any given trip, it is not clear whether a more-exploratory vessel would have higher or lower costs. For example, assume Vessel A is more exploratory as defined by visiting more sites on a trip than Vessel B. If Vessel B goes to one site but this site is further from port than the sites that Vessel A visits, then it is possible that Vessel B incurred more travel costs than Vessel A even though she only visited one site on a given trip. Because of this ambiguity, we focus our main analysis on revenues. However, we have now included an auxiliary analysis in the SI looking at the correlations between entropy and travel costs. Specifically, Fig S7 (Supplementary Information and below) shows the relationship

between choice entropy and revenue per km travelled. Distance travelled is a commonly used proxy for travel costs especially when vessels are of similar technology. We find that higher entropy vessels have lower revenue per km travelled. This does not imply that there is no economic advantage to being a less-exploratory vessel. Recall that we found that in the stable environment that there was no comparative advantage in being one type or another on average.

Fig. S7 Investment of resources in exploration likely requires additional travel costs relative to an exploitation strategy. To investigate whether vessels with higher choice-entropy EETO strategies incurred higher costs relative to revenues, we regress entropy score, S_{μ} , on revenue per kilometer travelled. We find that higher entropy vessels travelled farther per unit of revenue ($F_{[104]} = 17.29$, $P < 0.001$), confirming that there are measurable costs to investment in exploration.

3. Grid size may affect their results. The authors utilized a 30x30 grid. Now, the authors did vary the size of this grid by +/- 10%. This is good, but waaaaay more exploration of how the grid size changes the results would be advised. So I would test 3 orders of magnitude (e.g., 3x3, 30x30, 300x300, although 3x3 makes very little sense, so maybe 30x30, 300x300, and 3000x3000). This will take a long time to analyze, but the methods should be the same, so it is just computer time. The concern is that with the wrong size grid, what looks like exploitative strategy will in fact be exploratory.

As the Reviewer notes, we did address this in our paper but we agree we it would be interesting to push the analysis further. As a point of clarification, we emphasize that we do not attempt to label any strategy as being categorically 'exploratory' or 'exploitative', but rather to compare strategies among boats and determine that one boat is on average *more* or *less* exploratory than another. To do so, we use the same grid for all boats and so, in principle, changing the resolution should not profoundly change the relative scores of the boats. For example, if Vessel A is less exploratory than Vessel B, then Vessel A will have a lower entropy score. When we increase the grid resolution, it is correct that Vessel A's entropy score will increase as cells become subdivided into smaller cells. However, since all vessels are analyzed with the same grid, Vessel B's entropy score will *also* increase by a similar magnitude, and thus Vessel B will always have a higher entropy score than Vessel A and in our framework would always be considered more exploratory.

To test whether this assumption is correct, in the revised manuscript we recalculate our entropy measure across a broad range of grid sizes, from 20 x 20 cells (= 400, less than half the current resolution of 900 cells) to 95x95 cells (= 9025, more than a full order of magnitude greater than the current resolution). We then use Spearman rank correlation tests to quantify the pairwise correlation between the scores for each vessel across the range of resolutions. A Spearman score (*rho*) of 1 means that the variables are perfectly correlated, and correlation decreases as rho tends towards -1 (meaning the variables are diametrically opposed). As can be seen in the revised Fig S2 (below), correlations between our study resolution (dashed vertical line) and all others are extremely high (mean rho = 0.96; SD = 0.016) and all tests are highly significant ($P < 0.001$). We emphasize that this sensitivity analysis is not to determine the "correct" resolution for our study as there is no objective answer to that, but rather to demonstrate that changing the grid size by over an order of magnitude has a limited effect on our calculations.

Fig. S2 To test the sensitivity of our results to the chosen raster size (30 x 30 = 900 cells), we recalculate choice entropy using grids ranging from 20 x 20 = 400 to 95 x 95 = 9025 cells, which represents more than an order of magnitude difference in scale. Blue markers show Spearman correlation scores (*rho*) between our chosen resolution (dashed vertical line) and the other resolutions. Correlation is extremely high (mean rho = 0.96; sd = 0.02; for all tests, $P < 0.001$) between our chosen resolution and all others, indicating that our calculations are not particularly sensitive to raster size.

In addition, I wonder about a different sort of analysis. Is there a continuous variable that could be used, such as total distance traveled? You could just integrate over path length and see if that is influenced by their covariates. Discretizing is fine; the concern is that they overlook other behaviors that truly reflect exploration. For example, in rats, exploration often results in more turning. Do the boats turn more? Questions like that.

Apologies - we may not have been as clear on this point as we could have been. To identify foraging, we do indeed use continuous variables at an early stage in the analysis. As discussed in the Methods section, we leverage a dataset gathered by on-board fisheries observers who accompany a subset of fishing trips. The observers record the time when fishing gears are deployed and recovered. We cross-reference these records with the time-stamped vessel movement tracks from the observed vessel to label the segments of the tracks when the vessel is actively fishing. We then use

supervised learning to train a *random forest* classifier to identify this ‘foraging’ behavior from the characteristic movements of a vessel engaged in fishing, as encoded in continuous movement variables such as turning angles and speed as well as exogenous variables such as depth and time of day. Finally, we deploy our trained classifier on data from all the trips without an on-board observer, allowing us to identify where and when each vessel is fishing. In the manuscript, we have directed interested readers to a paper we published in ICES Journal of Marine Science where we go into a great deal of detail on the process.

Once we have identified where our agents are foraging, we *then* discretize space by creating vessel-specific portfolios of fishing grounds. Although we recognize that there are some caveats to discretizing data, it offers two important advantages for our purpose. First, portfolios of discrete locations allow us to ask whether a given location is being visited for the first time or is being revisited, which implicitly encodes the role of memory in the data and would be difficult with a continuous analysis. Second, fishers themselves discretize space into “fishing grounds”, often labelling these with site-specific nicknames. Although a boat’s movement is in continuous space, a decision to travel to a known fishing ground is really a discrete choice. So discretizing decisions more accurately models the underlying decision process in our system. For both of these reasons, models based on discretized space are commonly used in economics to model fishing behavior. For examples and links to the literature, please see:

- Sanchirico, J.N., Wilen, J.E., 1999. Bioeconomics of Spatial Exploitation in a Patchy Environment. *J. Environ. Econ. Manag.* 37, 129–150
- Smith, M.D., 2010. Toward an econometric foundation for marine ecosystem-based management. *Bull. Mar. Sci.* 86, 461–477
- van der Lee, A., Gillis, D.M., Comeau, P., Quinn, T., 2014. Comparative analysis of the spatial distribution of fishing effort contrasting ecological isodars and discrete choice models. *Can. J. Fish. Aquat. Sci.* 71, 141–150
- Vermard, Y., Marchal, P., Mahévas, S., Thébaud, O., 2008. A dynamic model of the Bay of Biscay pelagic fleet simulating fishing trip choice: the response to the closure of the European anchovy (*Engraulis encrasicolus*) fishery in 2005. *Can. J. Fish. Aquat. Sci.* 65, 2444–2453.

Note: Comment 4 has been moved to the start of our replies.

5. Please change ‘explorative’ to ‘exploratory’.

We debated this point ourselves. We decided to use ‘explorative’ as it was the semantic counterpoint to ‘exploitative’, there being no such word as ‘exploitatory’. However, we have no objection to the requested edit and we have changed the term throughout the text.

Note: Comment 6 has been moved to the start of our replies.

7. Pleased to note that the authors looked at the residuals for structure!

Thank you!

8. Regression notes:

a. I would include all the covariates in a single regression. Perhaps that was what in fact was done?

Yes, we did indeed include them all in a single regression prior to simplification by stepwise selection. We have now added the results of the full, unsimplified models to a new table, S2.

b. I would include weather, market price of fish, gasoline prices, and so forth in the regression as well.

We took a commonly used approach to handling these variables, which is to create a trip-level control variable using the revenue data of all other vessel fishing at the same time. The reasoning behind this approach is that these variables 1) have similar effects on all vessels fishing on a given day; and 2) are often difficult to measure directly, such as temporal variation in fish behavior which affects catch. We call the variable *contemporaneous fleet performance*, and it collapses these time-varying factors into a single proxy that allows us to compare fishing trips that took place at different times. Please see Fig. 3B in the manuscript for a scatterplot of the values over time.

c. Please use BIC as well as AIC to confirm best fit models.

Models have been refitted using BICs as well as AICs, and δ BIC values have now been added to Table S1.

d. More generally, there are a million covariates to control for. Run the kitchen-sink model. It sucks because it will eat up tons of your variance, but it is the right thing to do if you are going to use these models to infer effects. If the kitchen-sink model does not work, then you say that, and start doing fancier things—running power analyses, choosing more complex link functions, and the like.

There are certainly advantages to the “kitchen sink” approach, and we did include all of our covariates in a single model as the first exploratory step of model fitting. We have now added a new table, S2, which details the fully specified model estimates. As the Reviewer notes, however, there are also drawbacks to leaving redundant variables in a model, especially when relatively few replicates are available. In the particular case of our data, the most concerning drawback is that we have very noisy “real world” data from only 55 boats during the disturbance among which to partition variance. We have even fewer vessels to fit the model to in the first 10 days of the closure (Fig 3C in manuscript). For this reason, we adopted the classical data-driven approach of model simplification using stepwise deletions, which is common in the natural sciences. We found that none of the dropped variables were even marginally significant. In our revisions, we have added BICs to the model fitting process as requested and modified the presented results accordingly.

9. Is there really no work on removing bandits? Steyvers et al. have extensively studied different bandit problems—the authors should confirm this is the case.

We did not mean to imply that there is no work whatsoever on removing bandits but rather that this is not the norm in empirical human studies. Our existing text reads [“Although the machine-removal paradigm does not seem to be commonly employed in bandit trials...”]. However, to make the point clearer, we have modified the text by including the italicized words: [“Although the machine-removal paradigm does not seem to be commonly employed in bandit trials *involving human subjects...*”], which we believe to be correct.

10. Pleased to see that the data were curated! Good data is curated. However, please note in the text how many boats were excluded (and if there is room, why).

This information was somewhat buried in the Supplementary Information. To recap, a total of 125 longline vessels had data from the study period. However, 19 of these were subsequently excluded because the sparseness of their data caused erratic and unreliable convergence of the entropy metric, as can be seen in Fig S1, reproduced below. Collectively these vessels accounted for less than 5% of the logbook records. We have added the following text to direct the reader’s attention to Supplementary Information:

“The total fleet size of longline vessels active during our study period was 125, 19 of which were dropped for having sparse and erratic data (Fig. S1), leaving 106 vessels.”

Fig. S1 Deleted vessel records. Cumulative entropy trajectories for 19 vessels with statistically outlying trajectory slopes. Trajectories have been staggered horizontally to improve clarity. The horizontal axis shows the choice sequence and the vertical axis shows the entropy score. The erratic trajectories are caused by sparse data and the vessels were removed from the analysis.

11. The supplemental plots and main plots need more labelling (blue = X, green = Y, descriptions of the axes, etc.).

Done - SI figure labels have been revised.

Signed: David L Barack, Columbia University

Reviewer #2 (Remarks to the Author):

This paper reflects the improved data availability for people studying fisheries. The behaviour of individual boats can be followed as they go out on fishing trips. The study concerns long-line fisheries and seems to be both interesting and correct. My own studies concerned the groundfish fisheries of Nova Scotia. For these we developed a model of the fisheries which generated the movement of fleets and their resulting catches and profits. Interesting differences emerge between the fishing studied in this paper and the one we studied in Nova Scotia. We certainly raised the importance of explore/exploit behaviour. The main difference seems to be concerning the importance of 'social' factors. In Nova Scotia, fishing fleets were important rather than individual boats. Fishing affected the stocks quite rapidly and information of the spatial location of fish stocks was kept within fleets where possible. Coded messages and misleading information were used to stop knowledge spreading to rival fleets [This is fascinating behavior - Michele Barnes has found the same thing happening in Hawai'i: <https://www.pnas.org/content/113/23/6466>]. Fleets could have a statistical distribution of responses varying from completely random to all boats going to the best place they know of. Fleets could be composite – with some explorers and some exploiters. In case anyone was interested, publications were 1987 Allen P M and McGlade J M , "Modelling Complex Human Systems: A Fisheries Example", European Journal of Operational Research 30 (1987), pp 147-167.

I think that the paper is fine and offers an interesting analysis of this long line fishery, and the changing pay-off for exploration and exploitation. Data availability for individual boats opens up opportunities for new knowledge. The work and ideas described seems very clear and the conclusions seem correct. It should be published. Peter M Allen
Thank you for your review and your gracious comments, Dr Allen.

Reviewer #3 (Remarks to the Author):

The manuscript assesses real-world payoffs of contrasting explore/exploit trade-off (EETO) strategies using longline fishing in the Gulf of Mexico, leveraging a unique natural experiment where a large portion of prime fishing grounds was closed for five months in 2009 (due to excessive bycatches of endangered species of sea turtle) and many affected vessels were forced to relinquish habitually fished grounds and either fish elsewhere or retire during the closure. The authors assume that strategies investing too heavily towards either end of the spectrum are likely to be suboptimal and that before the disturbance, vessel-level performance may be maximized around intermediate strategies that more-evenly balance exploration and exploitation; however, during the disturbance, the more-explorative strategies should buffer against adverse impacts, when vessels with diversified portfolios of fishing grounds should benefit from their enhanced knowledge of non-impacted resources. The authors quantify EETO strategy using information entropy and develop cumulative entropy metric using the portfolio of fishing locations for each vessel. The findings show that in stochastic natural systems characterized by non-stationary rewards, the role of exploration in buffering against disturbance may be greater than its assumed role in improving performance.

This is a well-designed study with high quality data. The interpretation of the data is also well justified and the conclusions are novel and well supported by the results, and will be of interest to a wide readership. I recommend acceptance of the manuscript in Nature Communications essentially as is.

Thank you for taking the time to review our work.

I only have a few minor comments:

- Is 106 the total longline fleet in the US GoM or a sample? Perhaps a short description of fishery would be useful in order to calibrate the representativeness of the results.

Reviewer 1 raised a similar point in his comment #10. For succinctness, we do not reproduce our full reply here but we acknowledge that this information was obviously not clear in the original manuscript and we have added the following text to direct the reader's attention to Supplementary Information where the information is available:

"The total fleet size of longline vessels active during our study period was 125, 19 of which were dropped for having sparse and erratic data (Fig. S1), leaving 106 vessels"

- During the closure of the fishery, did the boats receive any subsidy or were they compensated in any way? If so, could this affect the disturbance level and strategy of the boats?

Great question - we should have clarified this in the text. For the turtle closure there was no compensation to affected operations. First an Emergency Closure was implemented because there were too many vessels deploying bottom longline gear and interacting with endangered sea turtles. The Amendment that followed included permanent time/area closures, gear restrictions and a longline endorsement with the explicit intent to cull the least efficient operators and cap the bottom longline fleet. In these kinds of closures that are based on management actions rather than, say, disasters such as the *BP Deepwater Horizon* oil spill, there is no compensation for Gulf of Mexico fleets. We have added the following text to the Introduction in case Nature Communications readers have similar concerns:

"As the intervention was a fisheries management action rather than an emergency closure such as an oil spill, no compensation was given to the longline fleet".

Reviewers' comments:

Reviewer #1 (Remarks to the Author):

I will elide a summary for this second set of comments.

Overall: 8.5/10

Recommendation: (very) minor R&R

The changes in the paper reflect a shift to a foraging focus, new findings about patch residence time, and various other more minor modifications. I note at the outset that the change in emphasis to a foraging framework has revealed new findings! That's great news! I do wish the authors could incorporate these foraging findings more robustly in the discussion, as I note below.

I have questions about the changed or added content but also some other questions. The questions about the changed or added content are mandatory; the new questions are optional, though I would be grateful for clarificatory responses! I divide these two sets of comments as 'old' vs. 'new'. 'Old' comments are those regarding changes made to the manuscript in light of my first round of comments. 'New' comments are new questions/concerns etc. raised. Only the 'old' comments should bear on the editorial decision. There was only a single 'new' substantive comment anyway.

Old comments:

1. Did patch residence times vary before/after disturbance? I understand that the before disturbance fit did not identify prt as a significant influence on mean performance, but neither did that fit identify entropy. And yet, there was a significant difference before vs after for entropy distributions of the vessels. So are there before vs after disturbance differences in prt? (Basically, a plot like 3B, but for prt.)
2. I noted your correlation of prt and entropy.
 - a. I'd like to see a plot of them against each other
 - b. If I understand this correctly, your p is -0.242 , so your $R^2 = 0.0625$ or so. That means as measures of exploratory behavior, entropy and prt are basically uncorrelated. That's excellent news. If true, you need to emphasize this: these are two largely independent measures of exploratory behavior. But I still want to see the plot.
3. Wowowow! Prt really did have an effect! Amazing!
 - a. Was prt computed for both before and after the disturbance separately? Did that change? I note that this may not be possible to assess because of the few records for after the disturbance.
 - b. How exactly was a patch defined? And how were the times calculated? For example, did you just take the mean of the ML-algorithm-identified fishing activity for each grid square? If so, then the grid square is the patch, and the time is computed from the average of the duration of fishing activity in that square, where the fishing activity is determined by the ML algorithm. At any rate, please state somewhere how this was computed.
 - c. Can you have a more fine grained model that does not just use the mean computed prt, but the actual observed prt's for each visit to a patch?
4. Need to add a sentence in the discussion about how you have two measures of exploration—entropy and prt—and how they are both differently informative about fishing exploration. And, finally, how both revealed something interesting about fishing behavior after the disturbance, and what that revelation was.

New comments:

1. No need to hyphenate 'more' and 'less'. So change 'more-exploratory' to 'more exploratory', etc.
2. Fig. 3B: add indicators for the mean of the entropy distributions and an asterisk to indicate significance.
3. Fig 1: "most used" or "most utilized", not "most-habituated"

4. I understand that more entropic vessels stayed during the disturbance, shifting the distribution to be more entropic. I also understand that entropies were computed for the two years prior to the disturbance. Were there any changes in entropy estimates from before to during the disturbance? That is, did some ships suddenly get more exploratory? Although I also understand that there are only a few records (or even only a single record for some vessels) from the disturbance period.

Signed,
David L Barack, Columbia University

Reviewer #3 (Remarks to the Author):

The comments have been satisfactorily dealt.

Responses to reviews of manuscript NCOMMS-18-36347

Title: Disturbance Modifies Payoffs in the Explore-Exploit Trade-Off

Authors: Shay O'Farrell, James N. Sanchirico, Orr Spiegel, Maxime Depalle, Alan Haynie, Steven A. Murawski, Larry Perruso and Andrew Strelcheck

- Reviewers' comments are in black.
- Our responses are in blue.
- Text reproduced from the manuscript is in square parenthesis.
- New text added to the manuscript is italicized.

Reviewers' comments:

Reviewer #1 (Remarks to the Author):

I will elide a summary for this second set of comments.

Overall: 8.5/10

Recommendation: (very) minor R&R

The changes in the paper reflect a shift to a foraging focus, new findings about patch residence time, and various other more minor modifications. I note at the outset that the change in emphasis to a foraging framework has revealed new findings! That's great news! I do wish the authors could incorporate these foraging findings more robustly in the discussion, as I note below.

I have questions about the changed or added content but also some other questions. The questions about the changed or added content are mandatory; the new questions are optional, though I would be grateful for clarificatory responses! I divide these two sets of comments as 'old' vs. 'new'. 'Old' comments are those regarding changes made to the manuscript in light of my first round of comments. 'New' comments are new questions/concerns etc. raised. Only the 'old' comments should bear on the editorial decision. There was only a single 'new' substantive comment anyway.

Old comments:

1. Did patch residence times vary before/after disturbance? I understand that the before disturbance fit did not identify prt as a significant influence on mean performance, but neither did that fit identify entropy. And yet, there was a significant difference before vs after for entropy distributions of the vessels. So are there before vs after disturbance differences in prt? (Basically, a plot like 3B, but for prt.)

Thank you for the suggestion as the result is exciting. We have added a new panel (Fig. 3C in revised manuscript and below) which shows that the mean PRT of the fleet did *not* change during the disturbance. This result contrasts notably with the observed increase in choice entropy (Fig 3B). In addition to adding the new figure panel, we have added text to the Discussion on the contrast between the PRT and entropy results – please see our reply to Comment #4.

2. I noted your correlation of prt and entropy.
a. I'd like to see a plot of them against each other

We have now included a new panel in the manuscript, reproduced below, showing the correlation between PRT and choice entropy.

Fig 2C. There is a weak negative correlation (-0.242; solid line) between choice entropy and patch residence time. Markers show the scores for each vessel expressed in standard deviations around zero means. Darker shading indicates higher pre-disturbance performance.

b. If I understand this correctly, your ρ is -0.242, so your $R^2 = 0.0625$ or so. That means as measures of exploratory behavior, entropy and prt are basically uncorrelated. That's excellent news. If true, you need to emphasize this: these are two largely independent measures of exploratory behavior. But I still want to see the plot.

We agree – exciting stuff. Plot added. Please see also our reply to Comment #4. The following text has been added to the Introduction:

[The correlation between PRT and choice entropy is weak (Pearson coefficient, -0.242; $P < 0.001$; Fig. 2C) indicating that these metrics capture different facets of exploratory behavior.]

3. Wowowow! Prt really did have an effect! Amazing!

Yes, this is a great result!

a. Was prt computed for both before and after the disturbance separately? Did that change? I note that this may not be possible to assess because of the few records for after the disturbance.

You are correct that PRT was not computed during the disturbance owing the small sample size. Both PRT and choice entropy were calculated over a two-year period prior to the disturbance. Please see the Methods section for further details.

b. How exactly was a patch defined? And how were the times calculated? For example, did you just take the mean of the ML-algorithm-identified fishing activity for each grid square? If so, then the grid square is the patch, and the time is computed from the average of the duration of fishing activity in that square, where the fishing activity is determined by the ML algorithm. At any rate, please state somewhere how this was computed.

Yes, your assumption is correct and we should have been more explicit. We have amended the following text in the Methods:

Old text: [To develop the PRT score for each vessel i , we use the vessel's VMS data from the pre-disturbance study period to calculate the mean number of hours vessel i spent at any given location before moving to the next location.]

New text: [To develop the PRT score for each vessel, we use its portfolio of fishing locations (grid squares) identified by the random forest classifier. Over the two-year pre-disturbance period, we calculate the mean time interval that each vessel remains within each fishing location before moving to a subsequent fishing location or returning to port.]

c. Can you have a more fine grained model that does not just use the mean computed prt, but the actual observed prt's for each visit to a patch?

This is interesting and would allow us to delve deeper into the Marginal Value Theorem but is really beyond the scope of the current study. For the purposes of our research question – *Do 'explorers' fare better than 'exploiters' during a system shock?* – we feel that the mean PRT is appropriate as a long-term measure of behavioural bias. However, a more nuanced investigation of PRT itself could certainly be of interest for a subsequent analysis.

4. Need to add a sentence in the discussion about how you have two measures of exploration—entropy and prt—and how they are both differently informative about fishing exploration. And, finally, how both revealed something interesting about fishing behavior after the disturbance, and what that revelation was.

In light of the revised findings, we have added a number of sentences to the Discussion, where we cast the results in the context of “fast vs slow” and “broad vs narrow” explorers. The particular revelation was that although both lower PRT

vessels (fast explorers) and higher entropy vessels (broad explorers) fared better during the disturbance, only higher entropy vessels were more likely to continue fishing. The result is both insightful and intuitive, and highlights the benefits of investigating multiple facets of exploration. Much of the new text, however, is discontinuous and is not readily reproduced here without losing its context, and we direct your attention to the revised manuscript itself. We have also modified the following sentences in the Introduction (new text is in italics) to introduce the fast/slow and broad/narrow paradigms into the manuscript prior to the Results:

[PRT quantifies the dilemma of whether to continue exploiting at the current location or to explore a new location in the hope of increasing the rate of reward (Charnov 1974), *which may be thought of as slow vs fast exploration (Verbeek et al. 1994; Sih et al 2004). We expect that some agents may be slower to explore, systematically remaining longer than others at a given location.* We calculate PRT as the mean duration that each vessel remains within a given patch before moving on. In contrast, choice entropy *measures broad vs narrow exploration*, and aims to quantify how much effort agents invest in gathering information about their environment.]

New comments:

1. No need to hyphenate 'more' and 'less'. So change 'more-exploratory' to 'more exploratory', etc.

Please allow us to clarify. We have used hyphens when the term is a compound adjective, such as "more-exploratory vessels" but we have not hyphenated the words in other contexts such as "*dissipating advantage may result from exploiters switching strategies to become more exploratory*". We feel that hyphenating compound adjectives is helpful to the reader but we have no objection to the Editor removing them.

2. Fig. 3B: add indicators for the mean of the entropy distributions and an asterisk to indicate significance.

Dashed lines have been added to both Fig. 3B and the new panel, Fig. 3C, to indicate means, and asterisk has been added to Fig 3B. The legend has been modified accordingly.

3. Fig 1: "most used" or "most utilized", not "most-habituated"

Done.

4. I understand that more entropic vessels stayed during the disturbance, shifting the distribution to be more entropic. I also understand that entropies were computed for the two years prior to the disturbance. Were there any changes in entropy estimates from before to during the disturbance? That is, did some ships suddenly get more exploratory? Although I also understand that there are only a few records (or even only a single record for some vessels) from the disturbance period.

As you intuit, there are not enough trips during the disturbance to recalculate entropy. As can be seen from Fig. 2B (reproduced below), the choice entropy trajectories require a long time-series to reach a plateau and provide reliable estimates. Please see the Methods for further details.

Signed,
David L Barack, Columbia University

Thank you once again for your improvements to the manuscript, Dr Barack. We would like add your name to the Acknowledgments, if you have no objection.

Reviewer #3 (Remarks to the Author):

The comments have been satisfactorily dealt.

Thank you for your time and your edits.

REVIEWERS' COMMENTS:

Reviewer #1 (Remarks to the Author):

I would be pleased to be included in the acknowledgements.

Accept.

David L Barack, Columbia